# Geranyl Diphosphate Synthase (CrtE) Inhibition Using Alendronate Enhances Isoprene Production in Recombinant *Synechococcus elongatus* UTEX 2973: A Step towards Isoprene Biorefinery

Indrajeet Yadav [1], Akhil Rautela [1], Agendra Gangwar [1], Vigya Kesari [2], Aditya K. Padhi [3] and Sanjay Kumar [1,*]

[1]  Biofuel Research Laboratory, School of Biochemical Engineering, IIT(BHU) Varanasi, Varanasi 221005, Uttar Pradesh, India
[2]  Department of Botany, Banaras Hindu University, Varanasi 221005, Uttar Pradesh, India
[3]  Laboratory for Computational Biology and Biomolecular Design, School of Biochemical Engineering, IIT(BHU) Varanasi, Varanasi 221005, Uttar Pradesh, India
[*]  Correspondence: sanjaykr.bce@iitbhu.ac.in

**Abstract:** A hemiterpene, isoprene, is commercially produced from crude oil refining processes. As a result of fossil fuel depletion, isoprene production process development is gaining attention from recombinant cyanobacteria and other microbial systems for its industrial and biofuel applications. In the present study, a fast-growing and $CO_2$-tolerant cyanobacteria, *Synechococcus elongatus* UTEX 2973, is engineered with *Pueraria montana* isoprene synthase (*IspS*) at neutral site I (NSI) in the genome of *S. elongatus* UTEX 2973. Furthermore, to enhance isoprene production a key enzyme (isopentenyl diphosphate isomerase, *IDI*) of the methyl-D-erythritol 4-phosphate (MEP) pathway is also overexpressed at neutral site III (NSIII). Wild-type and recombinant strains of *S. elongatus* UTEX 2973 (UTEX *IspS* and UTEX *IspS.IDI*) are studied for growth and isoprene production in the presence of an inducer (IPTG) and/or inhibitor (alendronate). Alendronate is used for the inhibition of geranyl diphosphate synthase (CrtE), downstream of the MEP pathway that catalyzes dimethylallyl diphosphate/isopentenyl pyrophosphate (DMAPP/IPP) condensation in the recombinant UTEX 2973 strains. The docking studies on SeCrtE (CrtE of *Synechcoccus elongatus* PCC 7942) and alendronate as an inhibitor have revealed that alendronate binds more tightly than IPP in the cavity of SeCrtE, with a higher number of intermolecular interactions and energy. The UTEX *IspS* strain has shown isoprene production below the limit of detection in the presence of an inducer and/or inhibitor; however, production studies using UTEX *IspS.IDI* showed a maximum production of 79.97 and 411.51 µg/g dry cell weight (DCW) in a single day in the presence of an inducer only and an inducer along with an inhibitor, respectively. The UTEX *IspS.IDI* strain produced 0.41 mg/g DCW of cumulative isoprene in the presence of an inducer and 1.92 mg/g DCW in the presence of an inducer as well as an inhibitor during six days of production. The yield improvement of isoprene is observed as being 4.7-fold by using the inhibition strategy, which is used for the first time in the recombinant cyanobacterial system. The average productivities of isoprene obtained from UTEX *IspS.IDI* are observed to be 2.8 µg/g DCW/h in the presence of an inducer and 13.35 µg/g DCW/h in the presence of an inducer as well as an inhibitor. This study provides a basis for the process development and yield improvement in isoprene production using a novel inhibition strategy in fast-growing recombinant cyanobacteria. Recombinant strains and metabolic pathway inhibition studies can be used in future attempts to photosynthetically produce hemiterpenes.

**Keywords:** isoprene; recombinant cyanobacteria; alendronate; isoprene biorefinery; isoprene synthase; *Synechococcus elongatus*

## 1. Introduction

Fossil-based fuel shortages and global warming concerns have prompted the necessity of developing sustainable processes that are mainly based on solar energy, wind energy, bioenergy, and hydro energy. The harvesting of solar energy and utilization of $CO_2$ to produce biofuels through photosynthesis is one of noteworthy achievements of nature that could also be a solution for the future production of energy and commodity chemicals [1]. Cyanobacteria have emerged as promising cellular factories for the sustainable production of carbon-neutral biofuels to mitigate problems caused by the excessive consumption of petroleum-based fuels [2]. In addition, cyanobacteria show a higher photosynthetic efficiency and growth rate than plants; they also have the potential to divert a large amount of absorbed carbon into the synthesis of biotechnologically important biofuels and chemical feedstocks [3]. Cyanobacteria are the preferred photosynthetic biocellular factories to produce biofuels and chemicals due to the ease of genetic manipulation. Efforts have been made to produce various biofuel-related metabolites, such as isoprene [4], $\alpha$-farnesene [5], ethanol [6], isobutanol [7], limonene [8], and alkane [9], using recombinant cyanobacterial systems. Here, we report enhanced isoprene production by a metabolic pathway inhibition strategy using phototrophic recombinant cyanobacteria.

Isoprene, a five-carbon isoprenoid molecule, is widely used as feedstock in the production of synthetic rubber, pesticides, lubricants, and adhesives [10]. Commercially, isoprene is produced from hydrocarbon mixtures during the steam cracking of crude oil in the petroleum refining industry [11]. The biological synthesis of isoprene occurs in a variety of plants, including *Eucalyptus globulus*, *Pueraria montana*, *Populus alba*, and *Populus canescens* in response to heat stress and drought [12]. The production of isoprene at a large scale by plants is not economically feasible due to the large land requirements and high costs of downstream processing [13]. Some attempts have been made for the development of an economically viable process for producing isoprene by using recombinant microbial systems—mainly bacteria, yeast, and cyanobacteria [14–17]; however, the sustainable and renewable production of isoprene, using $CO_2$ from recombinant cyanobacteria, could be an appropriate model production system [18,19]. Studies have also shown that isoprene can be a potential alternative to petroleum-based fuels due to its high energy density and low viscosity properties [19,20]. Recently, the hydrogenated isoprene dimers ($C_{10}H_{20}$ compounds—derivatives of cyclobutene, cyclohexane, and cyclooctane) from recombinant cyanobacteria have been characterized as ideal drop-in jet fuel [19,21].

Dimethylallyl diphosphate (DMAPP), the precursor of isoprene, is biologically synthesized by two main pathways, i.e., the methyl-D-erythritol 4-phosphate (MEP) and mevalonic acid (MVA) pathways [22]. Studies on metabolic pathway engineering showed that isoprene synthesis mainly depends on the intracellular concentrations of isopentenyl diphosphate (IPP), DMAPP, and the activities of isoprene synthase (*IspS*), isopentenyl diphosphate isomerase (*IDI*), and geranyl diphosphate synthase (CrtE) [18,23]. Archaea, animals, and some bacteria use the MVA pathway, which uses acetyl coenzyme A as the primary precursor, whereas bacteria, cyanobacteria, microalgae, and plants use the MEP pathway [18]. The initial step of isoprene synthesis by the MEP pathway utilizes pyruvate and glyceraldehyde 3-phosphate in a condensation reaction to form deoxy-D-xylulose 5-phosphate (DXP), which is catalyzed by deoxy-D-xylulose-5-phosphate synthase (DXS) (Figure 1). Following a series of reactions, two final products, DMAPP and IPP, are synthesized. DMAPP is used as a precursor for isoprene synthesis by *IspS* [23]. Although cyanobacteria lack the key enzyme *IspS* gene, they can be genetically modified with a plant-origin *IspS* gene and bottleneck gene overexpression for the sustainable production of isoprene by using $CO_2$ and light [1].

**Figure 1.** Methyl-D-erythritol 4-phosphate (MEP) pathway in recombinant cyanobacteria and the inhibition of the geranyl diphosphate synthase enzyme (CrtE) by alendronate and other bisphosphonates. The heterologously expressed genes *IspS* and *IDI* are marked in red. Abbreviations used (metabolites): G3P—glyceraldehyde 3-phosphate; DXP—deoxy-D-xylulose-5-phosphate; MEP—methyl-D-erythritol 4-phosphate; CDP-ME—diphosphocytidylyl methylerythritol; CDP-MEP—diphosphocytidylyl methylerythritol phosphate; MEcPP—methyl erythritol-2,4-cyclodiphosphate; HMBPP—hydroxymethylbutenyl diphosphate; DMAPP—dimethylallyl diphosphate; IPP—isopentenyl diphosphate; NADPH—nicotinamide adenine dinucleotide phosphate; CTP—cytidine triphosphate; PPi—diphosphate; ATP—adenosine triphosphate; ADP—adenosine diphosphate; CMP—cytidine monophosphate; $Fd_{red}$—ferredoxin reduced; and $Fd_{ox}$—ferredoxin oxidized. Enzymes: DXS—deoxy-D-xylulose-5-phosphate synthase; DXR—DXP reductoisomerase; IspD—CDP-ME synthase; IspE—CDP-ME kinase; IspF—ME-cPP synthase; IspG—HMBPP synthase; IspH—HMBPP reductase; IDI—isopentenyl diphosphate isomerase; and IspS—isoprene synthase.

The first successful instance of isoprene production in cyanobacteria was reported by Lindberg et al. [4]. They introduced the *IspS* gene from the *Pueraria montana* plant into model cyanobacteria, *Synechocystis* sp. PCC 6803 (PCC 6803), to make an isoprene-producing recombinant strain. Bentley et al. [24] expressed seven genes of the MVA pathway in PCC 6803, and a fed-batch closed culturing system was used for the enhanced production of isoprene [24]. In another study, the overexpression of an endogenous bottleneck enzyme, DXS, in combination with the heterologous enzyme *IspS* was studied in closed and open continuous systems for isoprene production [25]. The enhancement of isoprene productivity was observed as being 4.5 times higher in the continuous open system than the closed system [25].

To the best of our knowledge, no inhibition strategy has yet been used in a recombinant cyanobacterial system to enhance isoprene production. Metabolic studies on downstream MVA/MEP pathways in animal, human, and plant systems suggested that the farnesyl diphosphate synthase (FPPS) enzyme can be inhibited by alendronate and other bisphosphonates [26–28], which is similar to the geranyl diphosphate synthase (CrtE) of cyanobacteria; the inhibition of cyanobacterial CrtE could lead to improved production of isoprene (Figure 1). Bisphosphonates are bioisosteres of pyrophosphates and used as potent inhibitors of enzymes that act downstream of the MEP pathway as well as utilize substrates containing a pyrophosphate moiety, such as IPP [27]. The synthesis of various terpenes is carried out by the condensation of prenyl pyrophosphate (IPP and DMAPP) precursors catalyzed by the prenyl transferase enzyme [29]. The DMAPP acts as a priming molecule for the addition of IPP in a condensation reaction for the synthesis of various terpenoids, such as monoterpenes ($C_{10}$), sesquiterpenes ($C_{15}$), diterpenes ($C_{20}$), and tetraterpenes ($C_{40}$) [22,30].

In the present work, we have focused on the enhancement of isoprene production in a fast-growing cyanobacterial strain, *Synechococcus elongatus* UTEX 2973 (UTEX 2973), by using a genetic engineering approach in combination with the inhibition of the downstream of the MEP pathway reaction by an alendronate inhibitor. To accomplish the objective, the *IspS* and *IDI* genes are integrated at neutral site I (NSI) and neutral site III (NSIII) in the UTEX 2973 genome, respectively. We have performed further in silico studies to check the inhibition of the SeCrtE (CrtE of *S. elongatus* PCC 7942) enzyme by alendronate using molecular docking tools, and it catalyzes the DMAPP and IPP condensation in *S. elongatus* strains. An alendronate inhibitor and/or IPTG inducer were further applied during isoprene production by UTEX *IspS* and UTEX *IspS.IDI* recombinant strains in a closed cultivation system to see the effect on yield improvement. This study presents a first-hand explanation of the enhanced production of isoprene by the inhibition of the reactions downstream of the MEP pathway by alendronate in recombinant UTEX 2973 strains. The present findings will help to further the development of the emerging technology of isoprene biorefinery and address the challenge of scaling-up the process.

## 2. Materials and Methods

### 2.1. Strains and Culture Conditions

*Escherichia coli* DH5α (DE3) cells were routinely used for cloning and propagating plasmids. *E. coli* HB101 cells were used for triparental conjugation. All of the *E. coli* strains were cultivated and maintained in a Luria–Bertani (LB) broth medium for liquid cultures and LB agar for solid-state plate cultures at 37 °C in the presence of suitable antibiotics, where necessary [31]. A fast-growing cyanobacteria strain, UTEX 2973, was provided by Professor Pakrasi, Washington University (St. Louis, MI, USA) [32]. UTEX 2973 cells were grown and maintained in a liquid BG-11 medium or on solid BG-11 agar plates [32] at 38 °C in daylight fluorescent tubes that emit 100 μmol photon/$m^2$/s PAR (photosynthetically active radiation). Recombinant strains of UTEX 2973 produced by genetic modification were grown and maintained under the same conditions as mentioned above in the presence of suitable antibiotics.

### 2.2. Preparation of Plasmid Constructs

To obtain the isoprene synthase gene (*IspS*), plasmid pBA2SkIKmA2 (Addgene plasmid #39214) was purchased, which contains the codon-optimized *IspS* gene of the *Pueraria montana* plant [4]. Plasmid pAM2991 (Addgene plasmid #40248) was also purchased, which contains an upstream and downstream DNA segment of the neutral site I (NSI) region of the *S. elongatus* PCC 7942 (PCC 7942) genome. An advantage of using the pAM2991 plasmid was that it also contains a BOM sequence (oriT), which is essential for triparental conjugation. Another plasmid, pBbE1k-RFP (Addgene plasmid #35333), was purchased, which was genetically modified further for insertion into the NSIII site of the UTEX 2973 genome [33]. In addition, the conjugal plasmid pRL443 (Addgene plasmid #70261) and the

helper plasmid pRL623 (Addgene plasmid #58494), to be used in triparental conjugation, were also purchased [34].

A first vector was constructed, containing the *IspS* gene under the P$_{trc}$ promoter as well as upstream and downstream segments of the NSI region targeted for integration at the NSI in the genome of UTEX 2973. For this, the *IspS* gene was amplified using high-fidelity Phusion Plus DNA Polymerase (Thermo Fisher Scientific, Waltham, MA, USA) from a pBA2SkIKmA2 plasmid with gene-specific primers, with EcoRI and BamHI flanking recognition sequences at the 5′ and 3′ ends, respectively, and PCR conditions mentioned in Supplementary Tables S1 and S2 through the use of a Veriti Thermal Cycler (Applied Biosystems, Waltham, MA, USA). The amplified PCR product (*IspS* gene) was purified by a QIAquick gel extraction kit (Qiagen, Hilden, Germany) and cloned into pAM2991 between the restriction sites mentioned above. After ligation, the constructed resulting vector was named pAM2991-*IspS* (targeting integration at NSI) and transformed into *E. coli* DH5α cells. Transformed cells were selected in the presence of 50 µg/mL of spectinomycin and streptomycin, and insert verification was carried out by colony PCR as well as double digestion. The pAM2991-*IspS* plasmid was isolated from *E. coli* DH5α cells, sequenced for confirmation of the *IspS* gene, and further transformed into competent *E. coli* HB101 cells (first Cargo HB101 strain) for conjugation.

Another vector was constructed that contains the NSIII upstream region (NSIII′), the NSIII downstream region (NSIII″) fused with a BOM sequence, and *IDI* gene under the P$_{trc}$ promoter in pBbE1k-*RFP* targeted for integration at the NSIII site in the genome of UTEX 2973. For this, sequential addition and the screening of 912 bp NSIII′, 549 bp *IDI* gene, and 900 bp NSIII″ fused with a 141 bp BOM sequence in a pBbE1k-RFP plasmid were performed. The PCR amplifications of the NSIII´ segment, *IDI* gene, and NSIII″ segment were carried out using gene-specific primers and PCR conditions mentioned in Supplementary Tables S1 and S2. The DNA sequences NSIII′ (SpeI restriction site at both ends) and NSIII″ (AvrII and PciI restriction sites at the 5′ and 3′ ends) were amplified by PCR from the genomic DNA of UTEX 2973 using Phusion Plus DNA Polymerase. Target DNA products were gel purified. The purified NSIII´ segment and the pBbE1k-RFP plasmid were digested with the SpeI restriction enzyme, gel purified, and ligated. The ligated product, pBbE1k-RFP-NSIII´, was transformed into competent *E. coli* DH5α cells. Transformed cells were selected with kanamycin (50 µg/mL).

The *IDI* gene was amplified from the *E. coli* genome with gene-specific primers with an NdeI cut site at the 5′ end and a BamHI cut site at the 3′ end (Supplementary Tables S1 and S2). The pBbE1k-RFP-NSIII´ plasmid and the *IDI* gene were digested with the same restriction enzymes, gel purified, ligated, and transformed into competent *E. coli* DH5α cells. The NSIII″ sequence and the BOM site were fused by overlap extension PCR. The pBbE1k-IDI-NSIII′ plasmid and fused NSIII″ BOM sequence were digested with restriction enzymes PciI and AvrII. The digested plasmid and insert were gel purified and ligated. The ligated product (pBbE1k-*IDI*-NSIII) was transformed in *E. coli* DH5α cells (Supplementary Figure S1), and transformants were selected in the presence of kanamycin (50 µg/mL). The pBbE1k-*IDI*-NSIII plasmid was isolated from *E. coli* DH5α, sequenced for confirmation of the *IDI* gene as well as NSIII site, and transformed into competent *E. coli* HB101 cells (second Cargo HB101 strain) for conjugation. All of the cloned DNA segments were confirmed by colony PCR as well as by digestion of recombinant plasmids with referred restriction sites.

### 2.3. Conjugal Transfer of Plasmid into UTEX 2973

The triparental conjugation technique was performed to produce two recombinant strains of UTEX 2973 that carried the heterologous gene *IspS* (UTEX *IspS*) and *IspS* along with *IDI* (UTEX *IspS.IDI*), using a previously described method [35]. For this, a helper HB101 strain was prepared through transformation with the pRL443 and pRL623 plasmids, and the two Cargo HB101 strains were prepared through transformation with a vector—pAM2991-*IspS*/pBbE1k-*IDI*-NSIII, as mentioned earlier. To produce the first recombinant strain, UTEX *IspS*, wild-type UTEX 2973 cells and the first Cargo HB101 strain carrying

the vector (pAM2991-*IspS*) were used; however, to produce the second recombinant strain, UTEX *IspS.IDI*, the first recombinant strain, UTEX *IspS*, and the second Cargo HB101 strain carrying the vector, pBbE1k-*IDI*-NSIII, were used.

A fresh culture of UTEX 2973 was grown to $OD_{730}$ ~0.75 ± 0.05 in 50 mL of BG 11 medium. Cultures grown overnight of helper and cargo strains were centrifuged at $3000 \times g$ for 10 min at room temperature using a centrifuge (Thermo Fisher Scientific, Waltham, MA, USA). Pellets were washed twice with an LB broth to remove residual antibiotics and then resuspended in half the volume of the LB medium of the initial culture volume. The helper and cargo HB101 strains were mixed in a 1:1 ratio (450 μL of the helper strain and 450 μL of the cargo strain) and kept at room temperature. One milliliter of the UTEX 2973 culture was taken for each conjugation reaction and centrifuged at $1500 \times g$ for 10 min at room temperature. The supernatant was discarded, and the pellet was washed twice with BG11 medium and resuspended in the same. Washed UTEX 2973 cells, 900 μL, were mixed with previously combined helper and cargo strains in a microfuge tube and incubated in the dark at room temperature for 2 h. Mixed cells were centrifuged at $1500 \times g$ for 10 min at room temperature; 1.6 mL of supernatant was discarded and the pellet was resuspended in the remaining supernatant. Resuspended cells were spread onto a MF-Millipore 0.22 μm pore size membrane filter (Sigma-Aldrich, St. Louis, MI, USA) placed on BG11 plus 5% LB (*v/v*) agar plates without antibiotics and incubated at 38 °C for 24 h in light fluorescent tubes emitting 100 μmol photon/m$^2$/s PAR light intensity. After incubation, the membrane filter was transferred to fresh BG-11 agar plates containing 50 μg/mL of spectinomycin and streptomycin and incubated at 38 °C for 4–5 days in light (100 μmol photon/m$^2$/s). Transformed UTEX 2973 colonies were streaked on fresh BG-11 agar plates containing the abovementioned antibiotics. Transformed colonies were subsequently re-streaked 4 times for the proper segregation of DNA. The transformed colonies of the recombinant cyanobacterial strain were named UTEX *IspS*. A similar procedure was used to develop the recombinant strain UTEX *IspS.IDI*, using UTEX *IspS* instead of UTEX 2973, and selected positive transformants in the presence of kanamycin, spectinomycin, and streptomycin (50 μg/mL).

### 2.4. Genomic DNA Isolation and PCR Analysis of Recombinant Strains of UTEX 2973

Genomic DNA isolation was performed using the protocol described previously [36]. Genomic DNA was isolated from UTEX 2973 and recombinant strains of UTEX 2973 (UTEX *IspS* and UTEX *IspS.IDI*) using freshly grown cultures (50 mL; $OD_{730}$ ~0.75 ± 0.05) by centrifuging at $3000 \times g$ for 10 min at room temperature. Cell pellets were resuspended in 400 μL of lysis buffer (Urea 4M, Tris-HCl 0.2 M (pH of 7.4), NaCl 20 mM, EDTA 0.2 M, 50 μL proteinase K (20 mg/mL)) and incubated for 1 h at 55 °C. Furthermore, 1 mL of a prewarmed DNA extraction buffer (CTAB 3%, NaCl 1.4 M, EDTA 20 mM, Tris-HCL 0.5 M (pH of 8.0), sarkosyl 1%, and β-mercaptoethanol 1%) was added to the lysed solution. The mixture was divided into two parts, and equal volumes of chloroform and isoamyl alcohol (24:1) were mixed and centrifuged at $10,000 \times g$ for 5 min. The upper aqueous phase was taken in a new tube, and 2 volumes of absolute ethanol in addition to 0.1 volume of 3M sodium acetate (pH of 5.2) were added; the mixture was incubated at −80 °C for 1 h. After incubation, the mixture was centrifuged at $10,000 \times g$ for 10 min at 4 °C; the supernatant was discarded and the pellet was washed with 70% ethanol. The pellet was air-dried and resuspended in an elution buffer. Thus, isolated DNA samples were quantified by a NanoDrop One UV–Vis spectrophotometer (Thermo Fisher Scientific, Waltham, MA, USA) and used as templates to perform PCR for the integration of *IspS* and *IDI* genes within the genomic DNA of UTEX 2973 using appropriate primers and PCR conditions, mentioned in Supplementary Tables S1 and S2.

### 2.5. Growth Profile and Dry Cell Weight Determination

Growth profile studies on wild-type and recombinant strains of UTEX 2973 were performed in cotton-plugged serum bottles under ambient $CO_2$ conditions, a temperature

of 38 °C, continuous white light (100 µmol photon/m$^2$/s), and shaking at 180 rpm in a Orbitek shaker photo incubator (Scigenics Private Limited, Chennai, TN, India). Samples (1 mL) were taken every 24 h, and the absorbance was measured by using a Cary 60 UV–Vis spectrophotometer (Agilent Technologies, Santa Clara, CA, USA) at 730 nm. The dry cell weight (DCW) of recombinant strains and the wild-type strain of UTEX 2973 was determined by the method described previously [37].

### 2.6. Expression Analysis of IspS and the IDI Gene in Recombinant UTEX 2973 Strains through Semi-Quantitative RT-PCR

Semi-quantitative RT-PCR was performed with the method described earlier with slight modification [38]. For the expression analysis, UTEX 2973 and the recombinant strains (UTEX *IspS* and UTEX *IspS.IDI*) were grown to OD$_{730}$ ~0.75 $\pm$ 0.05 in 50 mL of BG-11 medium by using suitable antibiotics (25 µg/mL) under the same growth conditions as described earlier. Subsequently, cultures of the recombinant strains (UTEX *IspS* and UTEX *IspS.IDI*) were induced, with a final concentration of 1 mM isopropyl β-D-1-thiogalactopyranoside (IPTG) for 24 h. Another set of cultures of recombinant strains without any induction was also cultivated under similar growth conditions. For the total RNA extraction, 50 mL of UTEX 2973 and recombinant strain cultures with and without induction were harvested by centrifuging at 5000× *g* for 10 min at 4 °C after 24 h of induction; pellets were powdered in a precooled mortar and pestle in the presence of liquid nitrogen. Total RNA was isolated from the resulting cell pellet biomass by using RNeasy Plant Mini Kit (Qiagen, Hilden, Germany) according to the manufacturer's recommendations. The complete removal of genomic DNA traces was performed using on-column DNase I digestion. RNA quality and purity were measured using a NanoDrop spectrophotometer, and RNA integrity was checked by electrophoresis. cDNA was synthesized through the reverse transcription of 1 µg of total RNA using a GoScript$^{TM}$ cDNA synthesis kit (Promega, Madison, WI, USA) and random primers in a 20 µL reaction volume according to the manufacturer's instructions. The cDNAs were diluted with autoclaved distilled water and stored at −20 °C until further analysis. Gene-specific primers for *IspS* and *IDI* genes were designed with the NCBI primer design tool, targeting the coding region with an amplicon size of approximately 130 bp, and are listed in Supplementary Table S1. cDNA, 50 ng, was used as a template for PCR amplification carried out with gene-specific primers and PCR conditions described in Supplementary Tables S1 and S2. A housekeeping gene, the DNA-dependent RNA polymerase subunit alpha (*rpoA*), was used as an internal control. The PCR products were run on 2% agarose gel, visualized under the Gel Doc system, and the level of expression of genes was measured using ImageJ software.

### 2.7. Analysis of Protein by SDS-PAGE

Protein was extracted from samples of wild-type and recombinant UTEX 2973 strains and run on SDS-PAGE, as in a previously described protocol [39]. Fresh cultures of wild-type and recombinant UTEX 2973 strains (UTEX *IspS* and UTEX *IspS.IDI*) were grown in 50 mL of BG11 with a suitable antibiotic up to OD$_{730}$ ~0.75 $\pm$ 0.05 and induced with 1 mM IPTG. After 24 h of induction, cells were centrifuged at 3000× *g* for 10 min at 4 °C. Cell pellets were resuspended in 300 µL of Tris-HCl buffer (50 mM, pH of 8.0) supplemented with the protease inhibitor 1 mM phenylmethylsulphonyl fluoride (PMSF) (Sigma-Aldrich, St. Louis, MI, USA), 5% glycerol, and 1% Triton X-100. The resuspended samples were lysed by sonication at 50% intensity for 10 min (10 s on and 15 s off). The lysed cell samples were centrifuged at 16,000× *g* for 30 min. Supernatants were taken and mixed with 62 mM Tris-HCl (pH of 6.8), 1% SDS, 5% β-mercaptoethanol, and 10% glycerol in a 1:1 ratio. The protein contents of the diluted supernatant samples were determined by a Bradford assay [40]. Protein samples (10 µg) of each strain were run on 12% SDS-PAGE resolving gel and stained using the silver staining method.

### 2.8. Geranyl Diphosphate Synthase (CrtE) Inhibition Studies: Structure Preparation, Molecular Docking Simulations, and Analysis

The X-ray crystal structure of SeCrtE (PDB ID: 7MY7) was taken as the initial coordinate for the molecular docking of alendronate [29]. Before docking, the structure was prepared using Molecular Operating Environment (MOE), v2022.03. To evaluate the binding and energetics of alendronate and IPP, molecular docking simulations were carried out using CB-Dock2 [41,42]. CB-Dock2 is an improved protein–ligand docking method with cavity detection, ligand docking, and homologous template fitting features. CB-Dock2 computes (i) ligand binding sites using a curvature-based cavity detection method, followed by the (ii) identification of the center as well as size of the ligand binding site, (iii) adjusting the docking grid box size and then utilizing Autodock Vina for molecular docking. In addition, it integrates the template-based docking engine to improve the identification of the binding site and binding pose prediction of the ligand. To perform the docking, the MOE-prepared monomeric structure of apo SeCrtE was considered. Twenty cavities were generated per molecule (IPP and alendronate), and molecular docking was performed for each reaction. Template-based docking was performed using the crystal structure of SeCrtE C-term His-tag with IPP (PDB ID: 7MY6) as a reference. The docked poses were then used for interactions and energetic analyses.

To check the effect of alendronate on the activity of key enzymes, IspS, IspH, and IDI, the molecular docking of these enzymes was performed with their respective natural substrates and alendronate. The modeled AlphaFold structure of IspS (UniProt ID: Q6EJ97, AF-Q6EJ97-F1) from *Pueraria montana* was docked with DMAPP and alendronate to compare the binding site and binding energies. Similarly, the modeled AlphaFold structure of the IspH enzyme (UniProt ID: Q31S64, AF-Q31S64-F1) was docked with hydroxymethyl-butenyl diphosphate (HMBPP) and alendronate to compare the binding site and binding energy. Finally, as the crystal structure of IDI was available (PDB ID: 1NFZ), it was directly used for docking with IPP and alendronate to evaluate the binding and energetics.

### 2.9. Isoprene Production Conditions and Quantification

Isoprene production experiments were executed in a batch culture mode in sealed wheaten serum bottles (Sigma-Aldrich, St. Louis, MI, USA). Recombinant strains (UTEX *IspS* and UTEX *IspS.IDI*) were inoculated with an active culture and cultivated in cotton-plugged serum bottles in BG-11 media for 4 days till $OD_{730}$ reaches ~0.75. Cultures were then supplemented with sodium bicarbonate (50 mM), a 4-(2-hydroxyethyl)-1-piperazineethanesulfonic acid (HEPES) buffer (10 mM), IPTG (1mM), and/or the sodium salt of an alendronate inhibitor (25 µg/mL); cultures were sealed using aluminum crimp and teflon septa by a crimper (Sigma-Aldrich, St. Louis, MI, USA). The sealed serum bottles contain 120 mL of recombinant cyanobacterial cultures and 40 mL of gas headspace.

Gas samples of 1 mL were taken from the headspace of serum bottles by using a gas-tight syringe at 24 h intervals as well as analyzed by a Nucon gas chromatography system 5765 (Nucon Engineers Private Limited, New Delhi, India) through the use of a Porapack Q packed column (80/100) (Thermo Fisher Scientific, Waltham, MA, USA) and flame ionization detector (FID). Isoprene was quantified by using a standard curve prepared from the known concentrations of vaporized pure isoprene (Sigma-Aldrich, St. Louis, MI, USA). Nitrogen was used as a carrier gas with a flow rate of 40 mL/min. The initial oven temperature was kept at 50 °C for 2 min, with an increase of 15 °C/min until 120 °C, followed by 1 min of hold time. Furthermore, the oven temperature was raised to 200 °C, with a ramp temperature of 10 °C/min. The injector temperature was kept at 150 °C and the detector temperature was set at 280 °C [37].

## 3. Results and Discussions

### 3.1. Construction of Plasmids and Recombinant UTEX 2973 Strains

The *IspS* gene of the *Pueraria montana* plant was selected because it had shown efficient isoprene synthase activity in previously reported studies when expressed under a light-

regulated $P_{psbA2}$ promoter in PCC 6803 model cyanobacteria to produce isoprene [4,24,37]. We used the pAM2991 plasmid, previously made for the integration at the NSI site in the genome of PCC 7942 by adding an NSI´ upstream region and NSI″ downstream region to accomplish homologous recombination [43]. UTEX 2973 is a close relative of PCC 7942 and shares 99.8% of genome sequences [32]. The NSI upstream and downstream sequences are 100% identical, as checked by the NCBI nucleotide BLAST. The codon-optimized *IspS* gene, without the chloroplast transit peptide, was cloned between the EcoRI and BamHI restriction sites under the $P_{trc}$ promoter in the pAM2991 plasmid, forming pAM2991-*IspS* (Table 1). Next, the pBbE1k-*IDI*-NSIII plasmid construct was prepared, in which the *IDI* gene from *E. coli* DH5α was cloned under the $P_{trc}$ promoter (Table 1). The NSIII´ and NSIII´´ sequences from the UTEX 2973 genome were cloned at the upstream and downstream regions of the *IDI* gene, respectively. The UTEX *IspS* recombinant strain was developed by transforming (triparental conjugation) the pAM2991-*IspS* plasmid into UTEX 2973. The *IspS* gene, together with a spectinomycin gene, was integrated at the NSI site of the UTEX 2973 genome by double homologous recombination, forming the recombinant strain UTEX *IspS* (Figure 2A). Transformant cells were grown for several generations in the presence of a kanamycin antibiotic for the proper segregation of UTEX *IspS* DNA (DNA homoplasmy), resulting in the replacement of the DNA segment NSI with the *IspS* transgene. Similarly, the UTEX *IspS.IDI* strain was developed by transforming the pBbE1k-*IDI*-NSIII plasmid into the UTEX *IspS* strain, resulting in the integration of the *IDI* gene at NSIII into the UTEX *IspS* genome. A scheme of genomic DNA integrations of the *IspS* and *IDI* genes in the UTEX *Isps.IDI* strain is represented in Figure 2B. The genomic integration of the *IspS* and *IDI* genes was confirmed by a PCR analysis using genomic DNA of recombinant strains (Figure 2C). The cultivation scheme of recombinant strains of UTEX 2973 for growth and isoprene production is given in Figure 2D.

**Table 1.** List of plasmids and strains used in this study.

| Plasmids and Strains | Description | Antibiotic Resistance | Reference |
|---|---|---|---|
| **Plasmid** | | | |
| pBA2SkIKmA2 | Contains codon-optimized *IspS* gene for cyanobacteria | Kanamycin | [4] |
| pAM2991 | One-step cloning vector for overexpression between EcoRI and BamHI restriction sites | Spectinomycin | [43] |
| pRL443 | Conjugal plasmid suitable for the mobilization of cargo plasmids in cyanobacteria | Ampicillin | [34] |
| pRL623 | Helper plasmid for bacterial conjugal DNA transfer | Chloramphenicol | [34] |
| pBbE1k-*RFP* | Contains the *RFP* gene between NdeI and BamHI | Kanamycin | [33] |
| pAM2991-*IspS* | Derivative of pAM2991, *IspS* gene cloned between EcoRI and BamHI sites under the $P_{trc}$ promoter | Spectinomycin | This work |
| pBbE1k-*IDI*-NSIII | Made from pBbE1k-*RFP*, *IDI* gene cloned between NdeI and BamHI sites under the $P_{trc}$ promoter | Kanamycin | This work |
| **Strain** | | | |
| UTEX 2973 | Wild-type | None | [32] |
| UTEX 2973 *IspS* | The *IspS* gene integrated at the NSI site of the UTEX 2973 genome | Spectinomycin | This work |
| UTEX 2973 *IspS.IDI* | *IspS* and *IDI* genes integrated at the NSI and NSIII sites of the UTEX 2973 genome, respectively | Spectinomycin and kanamycin | This work |

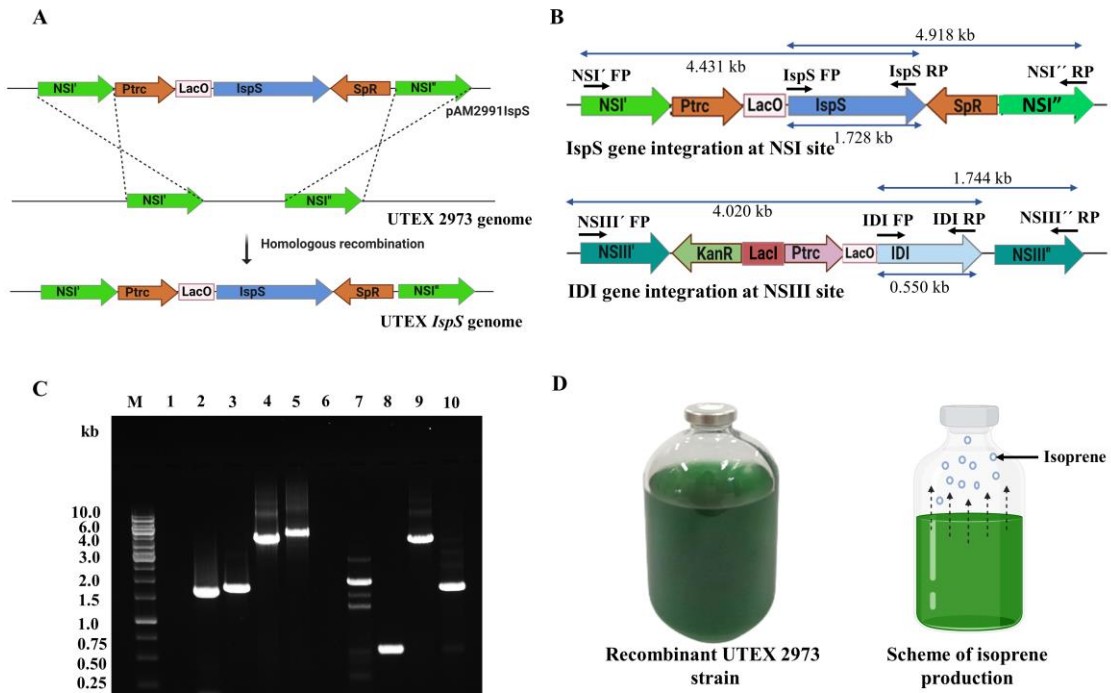

**Figure 2.** The homologous recombination process is used for the genomic DNA integration of the *IspS* and *IDI* genes at neutral site I (NSI) and neutral site III (NSIII), respectively. (**A**) Scheme of homologous recombination for *IspS* gene integration at the NSI site in the UTEX 2973 genome. (**B**) Schematic representation of the *IspS* and *IDI* genes integrated into the UTEX 2973 genome at the NSI and NSIII sites, respectively. $P_{trc}$—trc promoter; LacO—lac operator; SpR—spectinomycin resistance; KanR—kanamycin resistance; and LacI—lac repressor. (**C**) Gel image shows the genomic DNA integration of the *IspS* and *IDI* genes by PCR. The genomic DNA template of cyanobacterial strains, the primer pair used, and the amplified product size are as follows: M: DNA ladder; 1: primers—*IspS* FP and *IspS* RP, template—the UTEX 2973 genome, and no product was amplified; 2: primers—NSI´ FP and NSI´´ RP, template—the UTEX 2973 genome, and 1.6 kb of amplified product; 3: primers—*IspS* FP and *IspS* RP, template—the UTEX *IspS.IDI* genome, and 1.728 kb of amplified product; 4: primers—NSI´ FP and *IspS* RP, template—the UTEX *IspS.IDI* genome, and 4.431 kb of amplified product; 5: primers—*IspS* FP and NSI" RP, template—the UTEX *IspS.IDI* genome, and 4.918 kb of amplified product; 6: primers—*IDI* FP and *IDI* RP, template—the UTEX 2973 genome, and no product was amplified; 7: primers—NSIII´ FP and NSIII" RP, template—the UTEX 2973 genome, and 1.8 kb of amplified product; 8: primers—*IDI* FP and *IDI* RP, template—the UTEX *IspS.IDI* genome, and 0.550 kb of amplified product; 9: primers—NSIII' FP and IDI RP, template—the UTEX *IspS.IDI* genome, and 4.020 kb of amplified product; and 10: primers—IDI FP and NSIII" RP, template—the UTEX *IspS.IDI* genome, and 1.744 kb of amplified product. (**D**) Scheme of isoprene production by recombinant UTEX 2973 strains using serum bottle cultivation. Dashed arrows denote the shifting of isoprene molecules from the aqueous phase to gas phase in the sealed culture bottle.

### 3.2. Expression of IspS and IDI Transgenes and the Growth Profile of Recombinant UTEX 2973 Strains

The expression of heterologous genes, *IspS* and *IDI*, in recombinant strains, UTEX *IspS* and UTEX *IspS.IDI*, at the transcription level was determined by mRNA detection by using semi-quantitative RT-PCR (Figure 3A). Maximum levels of *IspS* and *IDI* transcripts have been reported in genetically engineered cyanobacteria at a 1 mM concentration of IPTG in a previous study [11]. To perform transcriptional analyses, UTEX 2973 and recombinant UTEX 2973 cultures (in the absence and presence of IPTG) were used for mRNA extraction and semiquantitative RT-PCR analysis. The results showed the presence of intense DNA bands in cultures supplemented with IPTG for the *IspS* gene in strains UTEX *IspS* and

UTEX *IspS.IDI*. Similarly, an intense band of the *IDI* gene was seen in the UTEX *IspS.IDI* strain (Figure 3A). However, faint DNA bands of *IspS* in the UTEX *IspS* and UTEX *IspS.IDI* strains, as well as *IDI* in the UTEX *IspS.IDI* strain, were observed in cultures grown without IPTG due to the leaky nature of the $P_{trc}$ promoter [44]. Since the UTEX 2973 strain does not have *IspS* and *IDI* (from *E. coli*) genes, no amplification was observed when using gene-specific semi-quantitative RT-PCR primers. Constitutively expressing housekeeping gene *rpoA* was used as an internal control, as has been suggested previously [45]. An amplified product of the *rpoA* gene was observed in strains UTEX 2973, UTEX *IspS*, and UTEX *IspS.IDI* (Figure 3A). Relative expression levels of the *IspS* and *IDI* genes were determined by using the *rpoA* gene as an internal control to normalize expression levels via densitometric analyses using ImageJ software. Normalized expression levels of the *IspS* gene were ascertained to be 0.20 (uninduced) and 0.9 (induced) in the UTEX *IspS* strain and 0.22 (uninduced) and 0.94 (induced) in the UTEX *IspS.IDI* strain; however, *IDI* gene expression levels were 0.15 (uninduced) and 0.93 (induced) in the UTEX *IspS.IDI* strain (Supplementary Figure S2). These findings suggest that the expression levels of the *IspS* and *IDI* genes are nearly equivalent to the internal control (*rpoA* gene) [45,46].

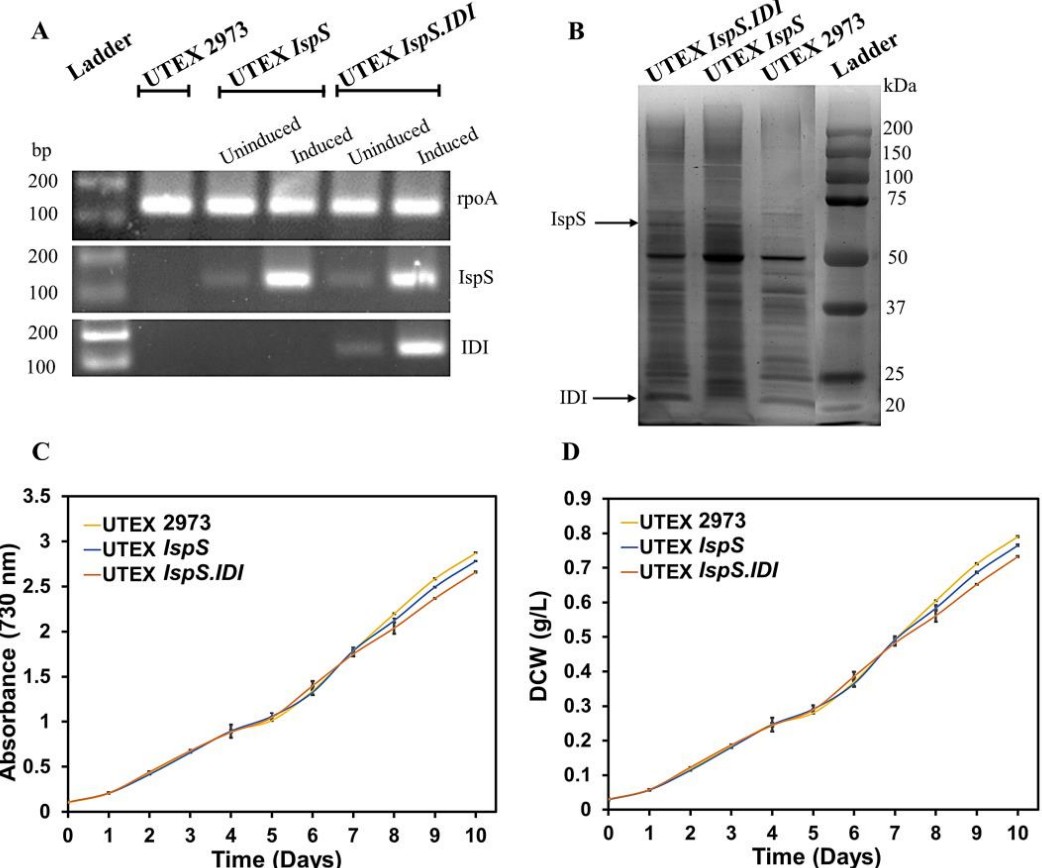

**Figure 3.** Gene expression analysis and growth profiles of the UTEX 2973, UTEX *IspS*, and UTEX *IspS.IDI* strains. (**A**) Semiquantitative RT-PCR: constitutively expressing the *rpoA* gene was used as a positive control, and gene-specific primers were used for the amplification of 130 bp DNA segments from the *IspS*, *IDI*, and *rpoA* coding regions. (**B**) Protein analysis by SDS-PAGE: a prominent band of IspS at 65.85 kDa was observed in the UTEX *IspS* and UTEX *IspS.IDI* strains, and a protein band of IDI at 20.5 kDa was observed in the UTEX *IspS.IDI* strain. (**C,D**) Growth profiles of the UTEX 2973, UTEX *IspS*, and UTEX *IspS.IDI* strains in cotton-plugged serum bottle cultures (100 μmol photons/m$^2$/s, 38 °C, ambient $CO_2$, and 180 rpm) expressed in absorbance (730 nm) and dry cell weight (DCW) (g/L).

Equal amounts of protein samples (10 μg) from the soluble fraction obtained from wild-type and recombinants of UTEX 2973 were run on SDS-PAGE and visualized under

Gel Doc. A prominent band of IspS of 65.85 kDa was observed in the UTEX *IspS* and UTEX *IspS.IDI* strains. Similarly, a 20.5 kDa protein band of IDI was observed in the UTEX *IspS.IDI* strain. No IspS and IDI protein bands were seen in UTEX 2973 (Figure 3B).

A growth study on UTEX 2973, UTEX *IspS*, and UTEX *IspS.IDI* was performed to see the effect of transgene expression on the growth characteristics of recombinant strains. Changes observed in the growth profiles of the recombinant strains were negligible when compared to the wild-type UTEX 2973 in the present study (Figure 3C,D). These findings are supported by a previous study in which the *IspS* and *IDI* genes were transformed into PCC 7942 without hampering the growth properties [12].

### 3.3. In Silico Studies on CrtE Enzyme Inhibition by Alendronate

Bisphosphonates are structural analogues of IPP and could inhibit IPP binding to DMAPP, blocking the downstream reactions to the MEP and MVA pathways that catalyze the synthesis of various terpenoid molecules [47–49]. Previously, phosphonates and bis-phosphonates have been used as potent inhibitors of human FPPS (hFPPS/CretE) to treat bone-related diseases, inhibit hFPPS for the development of pancreatic cancer drugs, and inhibit *Leishmania* and *Giardia* FPPS to inhibit cell proliferative activity. Alendronate has been shown to be one of the most effective drugs in such studies [27,50]. In the present work, using an in silico approach, we have reported the inhibition of the CrtE enzyme by alendronate to divert the engineered MEP pathway towards isoprene synthesis in recombinant UTEX 2973. The CrtE enzyme catalyzes three condensation reactions downstream of the MEP pathway making various terpenoids (Figure 1). Here, we have used the SeCrtE enzyme of PCC 7942 for docking studies to verify the activity of the alendronate inhibitor.

To evaluate the (i) binding site and interactions of alendronate and IPP in the binding pocket of SeCrtE, in addition to (ii) the difference in binding energies between alendronate and IPP with SeCrtE, docking studies were performed and the results were analyzed. First, alendronate was docked into the twenty identified cavities of SeCrtE, and it was found to be docked into a cavity with dimensions of 260 Å$^3$, center: 42,41,17 (x,y,z), and size: 18,18,18 (x,y,z). A high binding affinity of −4.7 (AuroDock Vina score) was obtained for the best pose. In contrast, when the SeCrtE structure was used for IPP docking, 20 cavities were identified into which the IPP was docked. It was found that IPP docked into one of the cavities in SeCrtE with an AutoDock Vina score of −3.2, having a cavity size of 79 Å$^3$, center: 38,45,23 (x,y,z) and size: 19,19,19 (x,y,z) (Figure 4). These results show that alendronate binds more tightly than IPP in the cavity of SeCrtE with a higher number of intermolecular interactions and energy (Table 2).

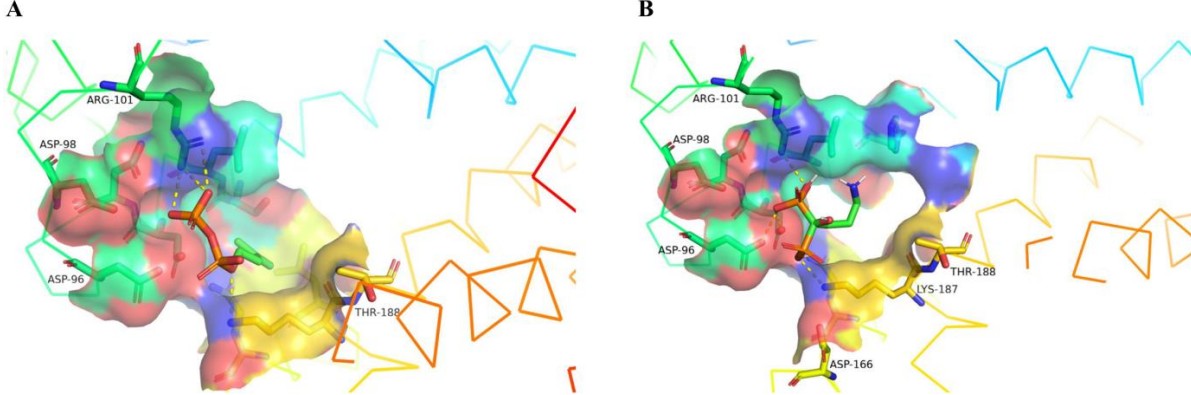

**Figure 4.** Docked pose of isopentenyl diphosphate (IPP) and alendronate in the cavity of SeCrtE. The surface line represented stereo views of docked poses of (**A**) IPP and (**B**) alendronate with SeCrtE. The hydrogen bonds have been represented as a dashed yellow stick, and the interacting residues are labeled, thus demonstrating higher binding in the case of alendronate as compared to IPP. SeCrtE—CrtE of *Synechococcus elongatus* PCC 7942.

**Table 2.** Types of intermolecular interactions between isopentenyl diphosphate (IPP) and SeCrtE as well as alendronate and SeCrtE, as obtained from docking simulations.

| Residue (SeCrtE) | Type of Interaction (IPP) | Residue (SeCrtE) | Types of Interaction (Alendronate) |
|---|---|---|---|
| Ser86 | Hydrogen bond | Ser86 | Hydrogen bond |
| Leu87 | Hydrophobic | Leu87 | Hydrophobic |
| Asp90 | Hydrophobic | Asp90 | Hydrophobic |
| Asp91 | Hydrogen bond | Asp91 | Hydrogen bond |
| Asp96 | Hydrogen bond | Asp96 | Hydrogen bond |
| Asp98 | Hydrogen bond | Asp98 | Hydrogen bond |
| Arg101 | Salt bridge | Arg101 | Salt bridge |
| Lys187 | Salt bridge | Leu159 | Hydrophobic |
| | | Gln163 | Hydrophobic |
| | | Lys187 | Salt bridge |

Other important MEP pathway enzymes, such as IspS, IspH, and IDI, were also checked for their molecular interaction and binding affinity with alendronate by using molecular docking studies. DMAPP, HMBPP, and IPP are the natural substrates of the IspS, IspH, and IDI enzymes, respectively. A high binding affinity of −6.5 (AutoDock Vina score) was obtained when IspS was docked with DMAPP, whereas a score of −5.0 was obtained when IspS was docked with alendronate. Similarly, a binding affinity of −4.1 was obtained when IspH was docked with HMBPP, whereas it was found to be −3.3 when docked with alendronate. Moreover, the binding affinity of IDI with IPP was found to be −4.2, as opposed to −3.4 when docked with alendronate. The results obtained in terms of binding energy and interacting sites indicate that alendronate does not inhibit the activity of these enzymes (Supplementary Figure S3). Thus, a comparative evaluation of the docking poses and scores of the CrtE, IspS, IspH, and IDI enzymes, with their respective substrates and alendronate, clearly demonstrates that alendronate significantly inhibits the binding of IPP to CrtE enzyme, whereas it does not affect the binding of DMAPP, HMBPP, and IPP with the IspS, IspH, and IDI enzymes, respectively.

*3.4. Isoprene Production by Recombinant UTEX 2973 Strains and the Effect of Alendronate on Production*

All of the production studies were performed in closed serum bottles. Culture bottles were incubated in an inverted position to avoid the evaporation of produced isoprene, as it is highly volatile. Isoprene concentration was measured in the gaseous phase of bottle headspace every 24 h after inducer and/or inhibitor addition. Previous studies have shown that isoprene and oxygen accumulation in a closed vessel inhibit cell growth and isoprene productivity [12,19]. Therefore, the headspaces of culture bottles were vented every 24 h to remove the accumulated isoprene and oxygen during the production phase. The isoprene level in recombinant strain UTEX *IspS* could not be quantified in the presence of IPTG, since the isoprene concentration was detected as being below the limit of quantification (LOQ) of the GC system (Figure 5A,B).

The IspS enzyme utilizes DMAPP as a substrate and converts it into isoprene, thus decreasing the intracellular concentration of DMAPP. Since IDI catalyzes the interconversion of IPP into DMAPP, we overexpressed the *IDI* gene in the UTEX *IspS.IDI* strain to increase the intracellular DMAPP flux towards isoprene production. We achieved a maximum isoprene value of 34.34 µg/L of culture, equivalent to 79.97 µg/g DCW of culture, in a single day, and a cumulative isoprene titer value of 0.16 mg/L of culture and 0.41 mg/g DCW in 6 days of cultivation for the *IspS* and *IDI* genes harboring the UTEX *IspS.IDI* strain in the presence of an inducer. The average isoprene production rate was calculated as being 1.11 µg/L/h, equivalent to 2.8 µg/g DCW/h, in 6 days with an inducer only (Figure 5C,D).

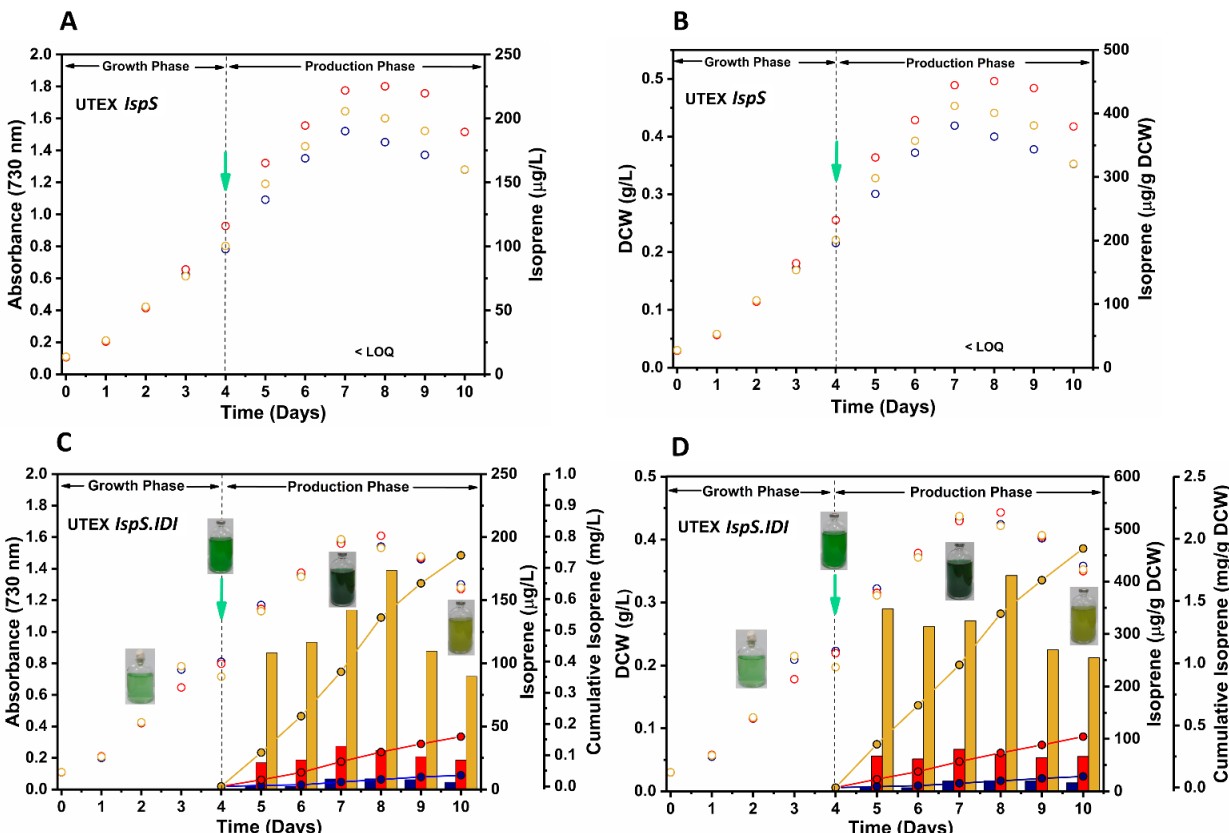

**Figure 5.** Growth and isoprene production (μg/L of culture broth) profiles of recombinant UTEX 2973 strains. (**A**,**B**) Strain UTEX *IspS*. (**C**,**D**) Strain UTEX *IspS.IDI*. Cotton-plugged cultures were grown (100 μmol photons/m$^2$/s, 38 °C, ambient $CO_2$, and 180 rpm) for 4 days. On the fourth day, cultures were supplemented with 50 mM $NaHCO_3$, 10 mM HEPES buffer, and an inducer (1 mM IPTG) and/or an inhibitor (25 μg/mL), after which they were sealed to entrap the produced isoprene in the bottle headspace. Cultures supplemented only with an inhibitor are depicted in blue, cultures supplemented only with IPTG are depicted in red, and cultures supplemented with IPTG and an inhibitor are depicted in orange. Hollow circles in the scatter plot show the growth profiles of recombinant UTEX 2973 strains (in presence of inhibitor- blue, IPTG- red, and IPTG plus inhibitor–orange), the bar diagrams show isoprene production, and the line plots show the cumulative isoprene production. The green arrow denotes the addition of an inducer and/or inhibitor. All data are expressed as the average values of duplicate experiments with <3% standard deviation. Limit of detection (LOD): 0.5 μg isoprene/L of the gas phase; limit of quantification (LOQ): 2.0 μg isoprene/L of the gas phase.

Downstream to the MEP pathway, the CrtE enzyme catalyzes the sequential condensation of 5-carbon allylic substrate DMAPP with three molecules of IPP, forming geranyl diphosphate (GPP), farnesyl diphosphate (FPP), and geranylgeranyl diphosphate (GGPP) in cyanobacteria [22,29]. These intermediates, GPP, FPP, and GGPP, are used for the synthesis of various monoterpenes, sesquiterpenes, diterpenes, and triterpenes in cyanobacteria [22]. In the first condensation reaction, IPP is added with DMAPP to form GPP. Here, we hypothesized that the inhibition of the CrtE enzyme can stop the synthesis of GPP, FPP, and GGPP in the MEP pathway. Thus, the accumulation of DMAPP will be available for isoprene synthesis by a heterologous expressed IspS enzyme. During the same time, free IPP will also be converted into DMAPP by the IDI enzyme, increasing the intracellular DMAPP concentration. For the first time, we reported the strategy of CrtE enzyme inhibition to enhance isoprene productivity by the alendronate inhibitor in a cyanobacterial system.

Isoprene production could also not be quantified in the presence of alendronate and IPTG by a UTEX *IspS* culture because the level of isoprene was below the LOQ, whereas the maximum isoprene level was achieved in this study by using the UTEX *IspS.IDI* strain in the presence of alendronate and IPTG. The maximum production observed was 173.62 µg/L of culture, equivalent to 411.51 µg/g DCW biomass (Figure 5C,D). In similar conditions, a cumulative isoprene amount of 0.74 mg/L of culture was achieved, equivalent to 1.92 mg/g DCW in 6 days, and productivity was found to be 5.14 µg/L/h, equivalent to 13.34 µg/g DCW/h (Figure 5C,D). When the UTEX *IspS.IDI* strain was cultured only in the presence of 25 µg/mL inhibitor without IPTG, a maximum isoprene amount of 8.50 µg/L, equivalent to 20.04 µg/g DCW, and a cumulative isoprene amount of 0.04 mg/L was achieved in 6 days, with an average production rate of 0.25 µg/L/h, equivalent to 0.63 µg/g DCW/h (Figure 5C,D). Isoprene production in cultures without IPTG induction by the UTEX *IspS.IDI* strain supports the leaky nature of the $P_{trc}$ promoter [44].

Studies reporting on the isoprene production from $CO_2$ when using recombinant cyanobacteria in closed cultivation and open continuous culture systems have been previously performed. Pioneering work reported 50 µg/g DCW isoprene production per day by integrating a codon-optimized *IspS* gene under a $P_{psbA2}$ promoter in PCC 6803 from the *Pueraria montana* plant [4]. This study was followed by several researchers utilizing different engineering strategies and culture conditions to enhance isoprene yields (Table 3). In our study, using an alendronate inhibitor in a UTEX *IspS.IDI* strain harboring *IspS* from *Pueraria montana* and *IDI* from *E coli* DH5α, we surpassed isoprene production levels in a closed system from previously reported isoprene production studies (Table 3). We found a cumulative isoprene yield of 0.41 mg/g DCW when induced only with IPTG. The yield was enhanced to 1.92 mg/g DCW when the culture was supplemented with an alendronate inhibitor; a 4.7-fold increase in isoprene yield was achieved. Since closed system cultivation of recombinant strain accumulates the isoprene and oxygen in the headspace, causing a negative impact on cell growth and isoprene production. Isoprene production levels could be further enhanced by culturing in an open system through continuous bubbling with air or $CO_2$, as has been reported in previous studies [12,19].

**Table 3.** Comparison of cumulative isoprene production and volumetric productivity in this study with previously reported studies.

| Cyanobacterial Strain | Engineering Strategy | Production Conditions | Cumulative Isoprene Production (Production Studies in Days) | Volumetric Productivity (µg/L/h) | Reference |
|---|---|---|---|---|---|
| *Synechocystis* sp. PCC 6803 | *IspS* gene integration under native $P_{psbA2}$ promoter in host genome | Closed-vessel cultivation | 50 µg/g DCW (1 day) | – | [4] |
| *Synechocystis* sp. PCC 6803 | *IspS* gene integration under native $P_{psbA2}$ promoter in host genome | Fed-batch cultivation in a closed system | 150 µg/L (8 days) | 0.78 | [37] |
| *Synechocystis* sp. PCC 6803 | *IspS* and MVA pathway enzymes were expressed | Fed-batch cultivation in a closed system | 250 µg/g DCW (8 days) | 1.53 | [24] |
| *Synechocystis* sp. PCC 6803 | *IspS* expressed under $P_{psbA2}$ promoter | Closed cultivation system, alkaline, and saline conditions | 120 µg/L (4 days) | 1.25 | [51] |
| *Synechocystis* sp. PCC 6803 | *IspS* gene inserted in pVZ325 replicative plasmid under various promoters | Closed and open cultivation systems, plasmid-based expression | 93 µg/g DCW (1 day; closed system) 336 µg/g DCW (1 day; open system) | 1.2 4.2 | [25] |
| *Synechococcus elongatus* PCC 7942 | *IspS, DXS, IspG,* and *IDI* genes were cloned under $P_{psbA2}$, $P_{trc}$, and $P_{cpcB}$ promoters | Open cultivation system, aerated with 5% $CO_2$ | 1260 mg/L (21 days) | 2500 | [12] |

**Table 3.** *Cont.*

| Cyanobacterial Strain | Engineering Strategy | Production Conditions | Cumulative Isoprene Production (Production Studies in Days) | Volumetric Productivity ($\mu$g/L/h) | Reference |
|---|---|---|---|---|---|
| *Synechocystis* sp. PCC 6803 | *IspS*, *IDI*, and *DXS* gene integration | Open cultivation system | 1.60 mg/L (4 days) | 16.6 | [19] |
| *Synechococcus elongatus* UTEX 2973 | *IspS* and *IDI* genes integrated under $P_{trc}$ promoter in cyanobacterial genome | Closed cultivation system, supplemented with alendronate as an inhibitor | 1920 $\mu$g/g DCW or 740 $\mu$g/L (6 days) | 5.14 | This work |

The need of today is to develop a circular-loop and efficient valorization-based cyanobacteria biorefinery by using considerations of process sustainability and economic viability. Techno-economic analyses of most of cyanobacteria-based biorefineries face challenges in regard to meeting economic feasibility and efficient bio-valorization due to expensive downstream processes, high input cost investment, and the production of limited products [12]. Furthermore, the self-shading in standard photobioreactor systems is one of the significant challenges to achieving high cell density [52]. To overcome this, the thickness of photobioreactors has to be curtailed. In one such study, Saccardo and colleagues made photobioreactors with thicknesses from 2 to 35 mm and observed strong self-shading at thicknesses above 8 mm [52]. Rapid growth rate and high amounts of cyanobacterial biomass accumulation are necessary to achieve the target volumetric productivity. UTEX 2973 was reported to have a high growth rate, high optimum growth temperature (38 °C to 42 °C), and high $CO_2$ assimilation rate, meaning that it could be a suitable photosynthetic cell factory for isoprene production [32]. Previous and present isoprene production studies have indicated that isoprene can be produced photosynthetically by recombinant cyanobacteria in a sustainable manner at a large scale [12,19]. Low downstream costs and higher biomass production are the main cost-reducing strategies for successful biorefineries. The continuous $CO_2$ fed operation in photobioreactors has shown maximum productivities for isoprene production by a recombinant cyanobacterial system rather than a closed-batch system [11,12,19]. Isoprene, being a volatile molecule (boiling point of 34 °C), is easily evaporated from photobioreactors. The recovery of isoprene has been reported as mainly being through two methods in continuous $CO_2$ fed mode photobioreactor operation [11,19].

In the first method, isoprene recovery is performed by using heptane as a solvent, maintaining its temperature at −40 C by using acetonitrile and dry ice [19]. In another recovery process, isoprene adsorbed on nonpolar adsorbers, eluted, and condensed at low temperatures (4 °C) [11]. A previous study suggests that activated charcoal could be a suitable and inexpensive option for isoprene adsorption and recovery when compared to the solvent recovery method at −40 °C [11]; however, the complete conversion of cyanobacterial biomass into biofuel and value-added products by a self-sustained biorefinery system is challenging. Multiple products, such as carotenoids, lipids, proteins, carbohydrates, nutraceuticals, and feedstocks, could be produced by the valorization of cyanobacterial biomass.

Figure 6 represents the flow diagram of an isoprene-based biorefinery consisting of various functional units, such as an isoprene production unit (consisting of a feed vessel and seed fermenter in which a cyanobacterial inoculum is prepared), isoprene recovery and processing unit, biomass recovery and valorization unit, anaerobic digestion unit, and wastewater treatment unit. Large-scale isoprene production can be performed by recombinant cyanobacteria in a photobioreactor using $CO_2$ from industrial flue gases. Isoprene production is carried out in a continuous photobioreactor system operated at optimum conditions of light, $CO_2$, and temperature. The $CO_2$ from the industrial flue gases could be used as a carbon source to maintain the sustainability and economic feasibility of the process. Isoprene vents out from the photobioreactor with the off gas and goes

to the isoprene recovery unit where it can be adsorbed on nonpolar adsorbents, eluted, and condensed to liquid form. Cyanobacterial biomass could be recovered in a biomass recovery unit and valorized into various value-added products. The generated wastewater could be treated in a treatment facility and reused in a production unit to develop a closed-loop-based biorefinery.

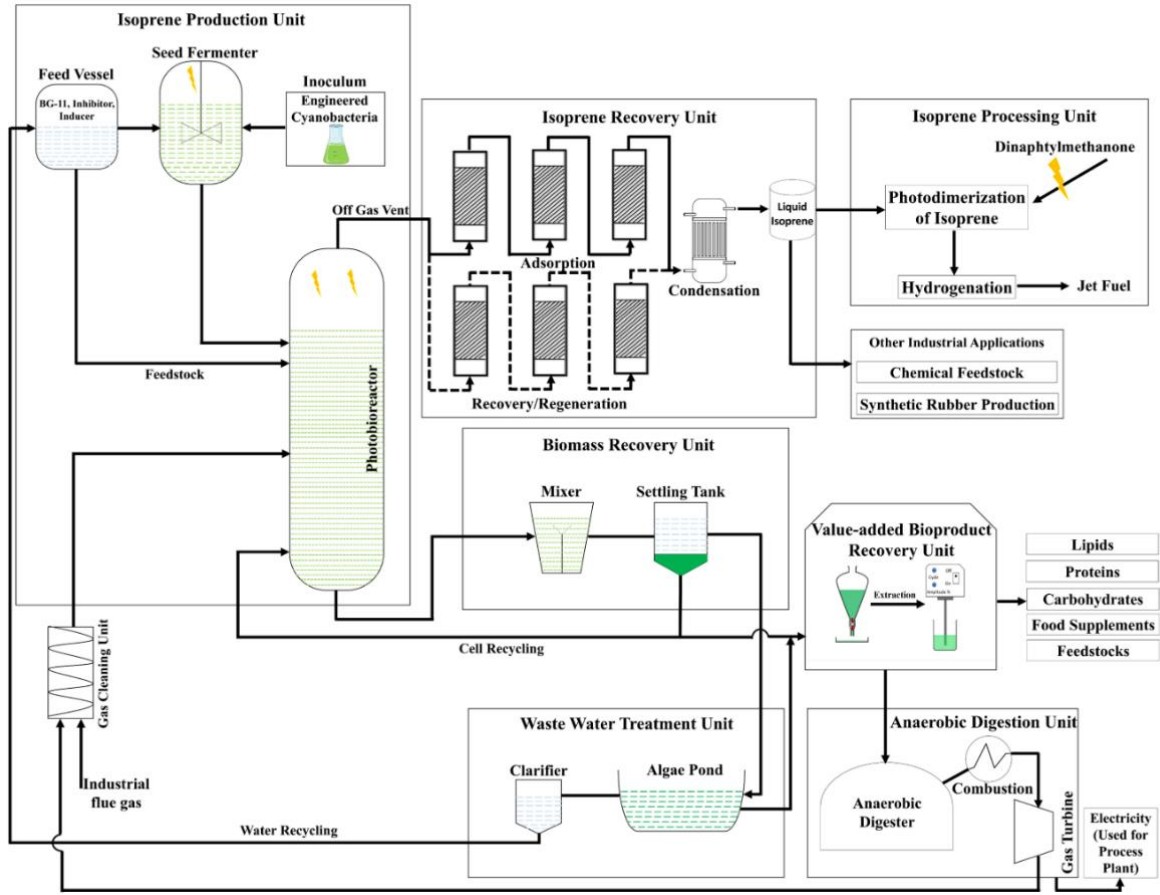

**Figure 6.** Process flow diagram for isoprene biorefinery concept with which to produce jet-category biofuel and value-added bioproducts using genetically modified cyanobacteria. Symbol ⚡ represents light illumination.

## 4. Conclusions

Isoprene, having the properties of advanced fuels, could be a potential alternative to fossil-based fuels. The sustainable production of isoprene using recombinant photosynthetic cyanobacteria has been our approach in this study. We engineered an MEP pathway by integrating the *IspS* gene from the *Pueraria montana* plant and the *IDI* gene from *E. Coli* DH5α under the control of a strong inducible promoter, $P_{trc}$, in the genome of UTEX 2973 at the NSI and NSIII sites, respectively. Recombinant UTEX 2973 strains were cultured in serum bottles containing BG11 media supplemented with 1 mM IPTG and 50 mM NaHCO$_3$ for isoprene production studies. Furthermore, we applied a CrtE enzyme inhibition strategy downstream to the MEP pathway by alendronate to enhance isoprene productivity. Docking studies on the CrtE enzyme with alendronate and IPP were carried out to compare molecular interactions and energetics. Alendronate was found to bind with the CrtE enzyme more tightly than IPP because of the higher binding affinity and intramolecular interactions between CrtE and alendronate. The findings of in silico studies were substantiated by the experimental application of alendronate for enhanced isoprene production by the recombinant UTEX 2973 strain. The UTEX *IspS.IDI* strain produced a cumulative isoprene amount of 0.41 mg/g DCW in 6 days in the presence of only IPTG.

When an alendronate inhibitor was supplemented along with IPTG, the recombinant UTEX *IspS.IDI* culture produced an isoprene amount of 1.92 mg/g DCW in 6 days, resulting in a 4.7-fold improvement in isoprene yield. This is the first study to report the use of alendronate to inhibit the CrtE enzyme, directing the carbon flux toward isoprene synthesis in recombinant cyanobacteria without affecting the activity of other key enzymes, such as IspS, IspH, and IDI. Thus, the recombinant UTEX 2973 *IspS.IDI* strain, being a fast-growing cyanobacteria, could be a potential photosynthetic cell factory for sustainable and improved isoprene production using an alendronate inhibitor. The findings of this work support the scheme of large-scale process development using recombinant cyanobacteria, a step towards scaling-up sustainable isoprene production.

**Supplementary Materials:** The following supporting information can be downloaded at: https://www.mdpi.com/article/10.3390/fermentation9030217/s1, Supplementary word file: Table S1, List of primers used; Table S2, Amplification of gene/DNA segment by PCR. Figure S1, Plasmid construct preparation and digestion verification of inserts. (A) Schematic representation of pAM2991-*IspS* construct preparation. (B) Scheme of sequential addition of DNA inserts to form a BbE1k-*IDI*-NSIII plasmid construct. (C) Digestion of pAM299-*IspS* to verify an *IspS* insert by EcoRI and BamHI, M—molecular marker, 1—plasmid digest. (D) Digestion of a pBbE1k-*RFP*-NSIII′ construct to verify an NSIII´ insert by the SpeI enzyme, M—molecular marker, 1—plasmid digest. (E) Digestion of a pBbE1k-*IDI*-NSIII´ construct to verify an *IDI* insert by the NdeI and BamHI enzymes, M—molecular marker, 1—*IDI* gene (positive control), and 2—plasmid digest. (F) Digestion of a pBbE1k-*IDI*-NSIII construct to verify an NSIII´´ insert by NdeI (an additional restriction site of NdeI is present in the BOM sequence), M—molecular marker, 1—plasmid digest. Figure S2, Densitometric analysis of semi-quantitative RT-PCR to test the expression levels of the *IspS*, and *IDI* genes in uninduced and induced (1 mM IPTG) conditions using ImageJ software (v.1.53t). The *rpoA* gene was used as an internal control to normalize the expression level. Figure S3, Lowest energy docked poses of the IDI, IspH, and IspS enzymes with their natural substrates and an alendronate inhibitor. (A) Docking poses of IDI with IPP and (B) alendronate. (C) Docking poses of IspH with HMBPP and (D) alendronate. (E) Docking poses of IspS with DMAPP and (F) alendronate. The substrates and alendronate have been labeled and shown as sticks.

**Author Contributions:** Conceptualization, I.Y., A.R. and S.K.; methodology, I.Y., A.R., A.G., V.K. and S.K.; software, S.K. and A.K.P.; formal analysis, I.Y., A.R., V.K., S.K. and A.K.P.; writing—original draft preparation, I.Y. and S.K.; writing—review and editing.; supervision, S.K. All authors have read and agreed to the published version of the manuscript.

**Funding:** This research was conducted using CPDA, IIT(BHU) Varanasi (2019-22), and the Research Support Grant Fund IIT(BHU) Varanasi (2018-21).

**Institutional Review Board Statement:** Not applicable.

**Informed Consent Statement:** Not applicable.

**Data Availability Statement:** Not applicable.

**Acknowledgments:** I.Y. and A.R. are grateful to the Ministry of Human Resource Development (MHRD), India, for financial aid. A.G. is thankful to the Department of Biotechnology (DBT), India, for providing research support. The authors would like to thank Hema Rajaram, MBD, BARC, India, for providing the *E. coli* HB101 strain and Himadri Pakrasi, Department of Biology, Washington University, USA, for providing the *S. elongatus* UTEX 2973 strain.

**Conflicts of Interest:** The authors declare no conflict of interest.

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
