# Peer review of "Geranyl Diphosphate Synthase (CrtE) Inhibition Using Alendronate Enhances Isoprene Production in Recombinant Synechococcus elongatus UTEX 2973: A Step towards Isoprene Biorefinery"

_fermentation, doi:10.3390/fermentation9030217_

Round 1

Reviewer 1 Report

The manuscript “Enhanced Production of Isoprene by Recombinant Synechococcus elongatus UTEX 2973 using Geranyl diphosphate synthase (CrtE) Inhibition Strategy: A Step Towards Isoprene Biorefinery” by Yadav et al. reports on phototrophic isoprene production with the fast growing cyanobacterial strain UTEX 2973. Two genes were introduced into this strain and overexpressed to enable isoprene synthesis from CO2, i.e. an isoprene synthase converting DMAPP into isoprene and isopentenyl diphosphate isomerase to foster the interconversion of IPP and DMAPP. Both modifications together indeed enabled also this strain to produce isoprene as shown before for other cyanobacterial strains. Interestingly, the production level could be increased almost 5-fold by applying alendronate as an inhibitor of the native IPP and DMAPP converting enzyme CrtE, which converters these isoprenoid precursors further to the precursors mono-, sesqui-, di-, and further multiterpenoids. This was achieved without observing a major effect on growth. This approach indeed shows big potential to improve terpenoid synthesis from CO2 by cyanobacteria and thus is considered to be of potentially high value for a future sustainable bioeconomy.

However, the paper suffers from severe language and formal deficiencies, which require major revisions. Content-wise, the manuscript – in my opinion – deserves publication after considering the comments listed below.

Major comments:

11. The manuscript is written in wordings often difficult to understand for the specialist and suffers from wrong or inappropriate use of articles, singular/plural, tense, and prepositions. This already becomes obvious in the title. It thus needs thorough possibly professional language editing.

22. A confirmation of the complete segregation of the constructed strains is missing, e.g. in Fig. 2. Primers before and after the inserts should be used. If “short” fragments (as in wildtype) still are detected (beside the target fragments), genome copies not carrying the inserts still persist.

33. An in silico inhibition study has been performed for CrtE. How is it with IspS, IDI, and possibly IspH accepting similar substrates as CrtE? Anything known/tested? This should be addressed.

44. A discussion section is missing and should be added. The section from line 542 to 577 (scheme 2 does not really exist) including Table 3 should be transferred to (be the core of) such a section and extended in terms of a more thorough comparison to other studies (represented in Table 3, volumetric productivities may also be considered) and the general performance of the UTEX strain. The authors not only introduced the inhibition concept, but also tested this special strain! So, what can be concluded in this respect?

Minor comments:

11. From the abstract, it does not become clear what the difference between maximum and cumulative isoprene levels are, making these statements a source of confusion.

22. Lines 45ff: biofuels are not the only alternative to fossil fuels (language).

33. Lines 49ff: cyanobacteria are promising but not (yet) ideal cell factories, as they suffer from rather slow growth and limited biomass concentrations…

44. Line 74: the “hydrogenated dimers” should be named.

55. Line 79: flux of IPP and DMAPP? A flux rather relates to a pathway (language)

66. Line 111: What came out of the cited study? This should be stated here.

77. Figure 1: MEP is not correctly depicted. The reaction catalyzed by IspG consumes 4 electrons (not 2) typically in the form of reduced ferredoxin and relieves one water molecule. The reaction catalyzed by IspH also relieves one water molecule. Geranylgeranyl diphosphate is the precursor of diterpenes and not of tetraterpenes.

88. Line 128 and elsewhere: literally there is no downstream MEP pathway. The MEP pathway ends at IPP/DMAPP. There are reactions downstream of the MEP pathway for terpenoid synthesis. (language)

99. For all chemicals, machines (shaker, GC, centrifuges), and devices (columns) used, the provider should be given, stating the company name, city, country (the state instead of the country in case of the USA). This largely is missing.

110.   Line 149: A reference should be given for the mentioned BG11 media.

111.   Line 166ff: The promoter used here should be mentioned

112.   Second and third paragraph on page 5: For primers the referencing of the supplemental material is missing.

113.   Line 341ff: The filling and headspace volumes should be given for the cultivation vessels used, especially for the serum bottles. Further, the time interval of headspace venting should be given, also in the results section.

114.   Table 1 should give the strains and plasmids under subtitles. Descriptions often are not clear and extensive enough (e.g., derivatives of what vector should be given).

115.   Figure 2 and text: Specifics for LacO (lac operator?) should be given. From the text, I understand that both genes (for IspS and IDI) are introduced under the control of the Ptrc promotor. So, is there any difference in their regulation? (see also minor comment 11)

116.   Line 517 (legend of Figure 5): should this not read 2 mg? 2µg would be very low for the upper quantification limit and not fit to the data given in Figure 5.

117.   Lines 561ff: As critical factors for cyanobacteria-based production, the rates / volumetric productivities should be addressed. UTEX has an advantage regarding growth/metabolic rates, enough? Volumetric productivities also depend on the cell concentration applied, which is inherently limited by self-shading in standard photobioreactor systems. So, such challenges should be included and discussed. Is adsorption really necessary? Would be a big cost factor.

118.   Part of the writing in Figure 6 is too small for appropriate representation.

Author Response

Response to the points raised by the reviewers

Manuscript ID: Fermentation- 2203704

The authors are thankful to the editor and reviewer for their valuable and constructive comments
and suggestions on the research article manuscript. The authors do agree that the points raised by reviewer are worthwhile incorporating to make the paper more meaningful, robust, and convincing. All other changes and additional information suggested by the reviewers have been accepted without any rebuttal and corrections have been made in the revised manuscript accordingly.

Reviewer 1

Major comment 1. The manuscript is written in wordings often difficult to understand for the specialist and suffers from wrong or inappropriate use of articles, singular/plural, tense, and prepositions. This already becomes obvious in the title. It thus needs thorough possibly professional language editing.

Response: The authors are thankful for the critical suggestions and comments of the reviewer for the manuscript. Comments and suggestions raised by the reviewer have been thoroughly rectified in the revised manuscript. Grammar mistakes and language improvisation were incorporated throughout the revised manuscript, where needed. Manuscript has been checked by native English speaker colleagues to improve its readability and language mistakes.

Major comment 2. A confirmation of the complete segregation of the constructed strains is missing, e.g., in Fig. 2. Primers before and after the inserts should be used. If “short” fragments (as in wildtype) still are detected (besides the target fragments), genome copies not carrying the inserts persist.

Response: Complete segregation in this study was performed by parts of PCR amplification using suitable primers and updated in Fig. 2 of the revised manuscript. The confirmation was performed by the genomic DNA of recombinant strains in the following manner. Forward primer (FP) of NSI’ to reverse primer (RP) of IspS and FP of IspS to RP of NSI". Forward primer (FP) of NSIII’ to reverse primer (RP) of IDI and FP of IDI to RP of NSIII”.

The amplicon size generated by using upstream and downstream neutral site primers was about 7.6 kb for integration at NSI and 5.2 kb for integration at NSIII. Although the amplification of FP NSI’ to RP NSI” and FP NSIII’ to RP NSIII” were tried using genomic DNA of recombinant strain in different PCR conditions and could not succeed. Therefore, PCR was done using the forward primer of the upstream neutral site and reverse primer of the gene of interest, and a separate PCR was done using the forward primer of the gene of interest and reverse primer of the downstream neutral site to show the transgene integration.

Reviewer concern about the small fragments of size NSI (wild) or NSIII (wild) were not observed in the PCR amplification for 7.6 kb for integration at NSI and 5.2 kb for integration at NSIII which could not be succeed.

Major comment 3. An in-silico inhibition study has been performed for CrtE. How is it with IspS, IDI, and possibly IspH accepting similar substrates as CrtE? Anything known/tested? This should be addressed.

Response: As suggested by the reviewer, docking of IspS, IDI and IspH was performed with alendronate and with their natural substrates DMAPP, IPP and HMBPP to compare the molecular interactions and binding affinities. The findings have been incorporated in the revised manuscript. Docking poses of enzymes with their substrate and alendronate have been demonstrated in Supplementary Figure S3.

Major comment 4. A discussion section is missing and should be added. The section from line 542 to 577 (Scheme 2 does not really exist) including Table 3 should be transferred to (be the core of) such a section and extended in terms of a more thorough comparison to other studies (represented in Table 3, volumetric productivities may also be considered) and the general performance of the UTEX strain. The authors not only introduced the inhibition concept but also tested this special strain! So, what can be concluded in this respect?

Response: The “discussion” heading was missing in the earlier manuscript and it has been added in the revised manuscript along with more discussion, which can be seen in the track change.

The suggested remark about productivity for table 3 has been added to the table in the revised manuscript.

The conclusion about the novelty of the inhibition concept and the speciality of the tested strain has been incorporated in the conclusion of the revised manuscript.

Minor comment 1. From the abstract, it does not become clear what the difference between maximum and cumulative isoprene levels are, making these statements a source of confusion.

Response: The raised ambiguity due to maximum and cumulative isoprene yields has been corrected in the abstract section of the revised manuscript.

Minor comment 2. Lines 45ff: biofuels are not the only alternative to fossil fuels (language).

Response: The language was rectified to make the sentence sensible which can be seen in the revised manuscript.

Minor comment 3. Lines 49ff: cyanobacteria are promising but not (yet) ideal cell factories, as they suffer from rather slow growth and limited biomass concentrations…

Response: As per the suggestion of the reviewer, the sentence has been modified in the revised manuscript.

Minor comment 4. Line 74: the “hydrogenated dimers” should be named.

Response: The hydrogenated dimers were named in the revised manuscript.

Minor comment 5. Line 79: flux of IPP and DMAPP? A flux rather relates to a pathway (language)

Response: The word flux was replaced with a concentration in the revised manuscript.

Minor comment 6. Line 111: What came out of the cited study? This should be stated here.

Response: The outcome of the cited study has been included in the revised manuscript.

Minor comment 7. Figure 1: MEP is not correctly depicted. The reaction catalyzed by IspG consumes 4 electrons (not 2) typically in the form of reduced ferredoxin and relieves one water molecule. The reaction catalyzed by IspH also relieves one water molecule. Geranylgeranyl diphosphate is the precursor of diterpenes and not of tetraterpenes.

Response: The concern about MEP metabolic pathway has been thoroughly studied and confirmed from the literature and KEGG metabolic pathway and necessary corrections have been made in figure 1 of the revised manuscript.

Minor comment 8. Line 128 and elsewhere: literally there is no downstream MEP pathway. The MEP pathway ends at IPP/DMAPP. There are reactions downstream of the MEP pathway for terpenoid synthesis. (language)

Response: The language of the sentence was modified as per the suggestion.

Minor comment 9. For all chemicals, machines (shaker, GC, centrifuges), and devices (columns) used, the provider should be given, stating the company name, city, country (the state instead of the country in case of the USA). This largely is missing.

Response: As suggested the provider of the machines and devices has been mentioned in the revised manuscript.

Minor comment 10. Line 149: A reference should be given for the mentioned BG11 media.

Response: The reference to the BG 11 media has been added to the revised version of the manuscript.

Minor comment 11. Line 166ff: The promoter used here should be mentioned

Response: In this study promoter used has been mentioned in the revised manuscript.

Minor comment 12. Second and third paragraph on page 5: For primers the referencing of the supplemental material is missing.

Response: Referencing for the primers of supplementary material has been done.

Minor comment 13. Line 341ff: The filling and headspace volumes should be given for the cultivation vessels used, especially for the serum bottles. Further, the time interval of headspace venting should be given, also in the results section.

Response: Culture and headspace volumes have been mentioned in the revised version of the manuscript. The time interval of headspace venting has been given.

Minor comment 14. Table 1 should give the strains and plasmids under subtitles. Descriptions often are not clear and extensive enough (e.g., derivatives of what vector should be given).

Response: As per the suggestion of the reviewer Table 1 has been elaborated with proper subtitles. The description column has been elaborated.

Minor comment 15. Figure 2 and text: Specifics for LacO (lac operator?) should be given. From the text, I understand that both genes (for IspS and IDI) are introduced under the control of the Ptrc promotor. So, is there any difference in their regulation? (See also minor comment 11)

 Response: LacO operator is present in both the plasmids constructed and is essential for the binding of LacI. Therefore, there is no difference in the regulation of both genes. Figure 2 has been modified accordingly.

Minor comment 16. Line 517 (legend of Figure 5): should this not read 2 mg? 2µg would be very low for the upper quantification limit and not fit to the data given in Figure 5.

Response: The data shown in figure 5 are in terms of µg of isoprene per liter of culture broth. Whereas LOD and LOQ values were determined from the vaporized isoprene in terms of µg of isoprene per liter of the gas phase by injecting 1 mL gas sample in GC. The mentioned LOQ and LOD are correct in the manuscript.

Minor comment 17. Lines 561ff: As critical factors for cyanobacteria-based production, the rates / volumetric productivities should be addressed. UTEX has an advantage regarding growth/metabolic rates, enough? Volumetric productivities also depend on the cell concentration applied, which is inherently limited by self-shading in standard photobioreactor systems. So, such challenges should be included and discussed. Is adsorption really necessary? Would be a big cost factor.

 Response: The authors agree with the suggestions of the reviewer Volumetric productivity of cyanobacterial production systems has been added. The growth rate of UTEX 2973 has been reported higher than other model cyanobacteria and is equivalent to heterotrophic yeast and some bacteria. The challenges related to the self-shading and recovery of isoprene and its cost of recovery added in the revised manuscript.

Minor comment 18. Part of the writing in Figure 6 is too small for appropriate representation.

 Response: The description of figure 6 has been elaborated in the revised manuscript.

A number of alterations have been brought into the revised manuscript. The authors have tried their best to address almost all the concerns of the Editor and the Reviewers point by point in the current version of the revised manuscript to make it improved and commendable for its publication as a research article in a peer-reviewed international journal “Fermentation journal” as well as to provide justified responses describing all the improvements that were brought in the revised manuscript.

Reviewer 2 Report

Line 61: Isoprene, a five-carbon isoprenoid molecule, is widely used as feedstock….. The comma after molecule is missing.

Line 76: Dimethylallyl diphosphate (DMAPP), the precursor of isoprene, is biologically… The comma after isoprene is missing.

Line 255: resuspended in 400 μLlysis buffer. I think there is a mistake here.

Line 279: Semi quantitative R-PCR was performed. It should be RT-PCR.

Line 349: 1. mL gas samples…. The full stop after 1 is unnecessary.

In general, sometimes it says ml and sometimes mL, always put the same. I prefer mL because you use µL.

Author Response

Response to the points raised by the Reviewer 2

Manuscript ID: Fermentation- 2203704

The authors are thankful to the editor and reviewer for their valuable and constructive comments
and suggestions on the research article manuscript. The authors do agree that the points raised by reviewer are worthwhile incorporating to make the paper more meaningful, robust, and convincing. All other changes and additional information suggested by the reviewers have been accepted without any rebuttal and corrections have been made in the revised manuscript accordingly.

Reviewer 2

Minor comment 1. Line 61: Isoprene, a five-carbon isoprenoid molecule, is widely used as feedstock….. The comma after molecule is missing.

 Response: The authors are thankful to the reviewer for suggesting valuable corrections. As suggested a comma (,) has been placed.

Minor comment 2. Line 76: Dimethylallyl diphosphate (DMAPP), the precursor of isoprene, is biologically… The comma after isoprene is missing.

Response: A comma (,) has been placed as suggested.

Minor comment 3. Line 255: resuspended in 400 μLlysis buffer. I think there is a mistake here.

Response: Sentenced modified to “resuspended in 400 μL of lysis buffer”

Minor comment 4. Line 279: Semi-quantitative R-PCR was performed. It should be RT-PCR.

Response: The sentence was corrected accordingly.

Minor comment 5. Line 349: 1. mL gas samples…. The full stop after 1 is unnecessary.

Response: The full stop was removed.

Minor comment 6. In general, sometimes it says ml, and sometimes mL, always put the same. I prefer mL because you use µL

Response: ‘ml’ was replaced with ‘mL’ throughout the manuscript

A number of alterations have been brought into the revised manuscript. The authors have tried their best to address almost all the concerns of the Editor and the Reviewers point by point in the current version of the revised manuscript to make it improved and commendable for its publication as a research article in a peer-reviewed international journal “Fermentation journal” as well as to provide justified responses describing all the improvements that were brought in the revised manuscript.

Reviewer 3 Report

Well established research with well presented results.

Author Response

Response to the points raised by the Reviewer 3

Manuscript ID: Fermentation- 2203704

The authors are thankful to the editor and reviewer for their valuable and constructive comments and suggestions on the research article manuscript. The authors do agree that the points raised by reviewers are worthwhile incorporating to make the paper more meaningful, robust, and convincing. All other changes and additional information suggested by the reviewer have been accepted without any rebuttal and corrections have been made in the revised manuscript accordingly.

Reviewer 3

Comment: Well-established research with well-presented results.

Response: The authors are very thankful for the positive and motivating words from the reviewer.
